# An updated nuclear-physics and multi-messenger astrophysics framework for binary neutron star mergers

Peter T. H. Pang [1,2], Tim Dietrich [3,4] ✉, Michael W. Coughlin [5], Mattia Bulla [6,7,8,9], Ingo Tews [10], Mouza Almualla [11], Tyler Barna [5], Ramodgwendé Weizmann Kiendrebeogo [12,13], Nina Kunert [3], Gargi Mansingh [5,14], Brandon Reed [5,15], Niharika Sravan [16], Andrew Toivonen [5], Sarah Antier [13], Robert O. VandenBerg [5], Jack Heinzel [17], Vsevolod Nedora [4], Pouyan Salehi [3], Ritwik Sharma [18], Rahul Somasundaram [10,19,20] & Chris Van Den Broeck [1,2]

The multi-messenger detection of the gravitational-wave signal GW170817, the corresponding kilonova AT2017gfo and the short gamma-ray burst GRB170817A, as well as the observed afterglow has delivered a scientific breakthrough. For an accurate interpretation of all these different messengers, one requires robust theoretical models that describe the emitted gravitational-wave, the electromagnetic emission, and dense matter reliably. In addition, one needs efficient and accurate computational tools to ensure a correct cross-correlation between the models and the observational data. For this purpose, we have developed the Nuclear-physics and Multi-Messenger Astrophysics framework NMMA. The code allows incorporation of nuclear-physics constraints at low densities as well as X-ray and radio observations of isolated neutron stars. In previous works, the NMMA code has allowed us to constrain the equation of state of supranuclear dense matter, to measure the Hubble constant, and to compare dense-matter physics probed in neutron-star mergers and in heavy-ion collisions, and to classify electromagnetic observations and perform model selection. Here, we show an extension of the NMMA code as a first attempt of analyzing the gravitational-wave signal, the kilonova, and the gamma-ray burst afterglow simultaneously. Incorporating all available information, we estimate the radius of a $1.4 M_\odot$ neutron star to be $R = 11.98^{+0.35}_{-0.40}$ km.

The study of the gravitational-wave (GW) and electromagnetic (EM) signals GW170817[1], AT2017gfo[2–12], and GRB170817A[13–15] has already enabled numerous scientific breakthroughs, for example, constraints on the properties of neutron stars (NSs) and the dense matter equation of state (EOS) at supranuclear densities[16–23], an independent measurement of the Hubble constant[22,24–28], the verified connection between binary NS (BNS) mergers and at least some of the observed

short gamma-ray bursts (GRBs)[29], and precise limits on the propagation speed of GWs[29]. These scientific achievements were enabled by the multi-messenger nature of GW170817.

Despite this enormous progress, results have been obtained by connecting constraints from individual messengers a posteriori, i.e., different messengers were analyzed individually and then combined within different multi-messenger frameworks to achieve the final

results. Such frameworks and attempts include, among others, the work of Breschi et al.[30] performing Bayesian inference and model selection on the kilonova AT2017gfo, Nicholl et al.[31] developing a framework for predicting kilonova and GRB afterglow lightcurves using information from GW signals as input, and the multi-messenger framework developed by Raaijmakers et al.[32]. Similarly, to these works, our previous Nuclear physics - Multi-Messenger Astrophysics (NMMA) framework has been successfully applied to provide constraints on the EOS of NS matter and on the Hubble constant[22,33], to investigate the nature of the compact binary merger GW190814[34], to provide techniques to search for kilonova transients[35], to classify observed EM transients such as GRB200826A[36], and to combine information from multi-messenger observations with data from nuclear-physics experiments such as heavy-ion collisions[23].

Here, we upgrade our framework to allow for a simultaneous analysis of kilonova, GRB afterglow, and GW data capitalizing on the multi-messenger nature of compact-binary mergers.

## Results

The full potential of our NMMA study becomes clear from Fig. 1 where we show a set of possible EOSs relating the pressure and baryon number density inside NSs. Different constraints can provide valuable information in different density regimes. For example, theoretical calculations of dense nuclear matter in the framework of chiral effective field theory (EFT)[37–41] or data extracted from nuclear-physics experiments, e.g., heavy-ion collisions[42] or the recent PREX-II experiment at Jefferson Laboratory[43], provide valuable input up to about twice the nuclear saturation density, $n_{sat} \approx 0.16$ fm$^{-3}$. GW signals emitted during the inspiral of a BNS or black-hole–NS (BHNS) systems contain information that probe the EOS at densities realized inside the individual NS components of the system, typically up to about five times $n_{sat}$, but the exact density range probed in such mergers depends noticeably on the mass of the component stars. Furthermore, radio observations of NSs can be used to infer their masses, e.g., by measuring Shapiro delay in a binary system. In particular, radio observations of heavy NSs with masses of about $2M_\odot$, such as PSR J0348+0432[44], PSR J1614-2230[45], and PSR J0740+6620[46], currently provide valuable information at larger densities than those probed by

inspiral GW signals. In addition, these observations provide a valuable lower bound on the maximum mass of NSs. Matter at the highest densities in the universe could be created in the postmerger phase of a BNS coalescence, i.e., after the collision of the two NSs in the binary. This phase of the binary merger might be observed through future GW detections with more sensitive detectors. Alternatively, this phase can be probed by analyzing EM signals connected to a BNS merger, i.e., the kilonovae, GRBs, and their afterglows. Finally, at asymptotically high densities that are not shown in the figure, the EOSs can be calculated in perturbative QCD[47] and might be used to constrain the NS EOS[48]. The combination of all these various pieces of information provides a unique tool to unravel the properties of matter at supranuclear densities.

### GW170817-AT2017gfo

With the NMMA framework, we analyze GW170817 simultaneously with the observed kilonova AT2017gfo. For the GW analysis, we have used the `IMRPhenomPv2_NRTidalv2` waveform model and analyzed the GW data obtained from the Gravitational Wave Open Science Center (GWOSC)[49] in a frequency range of 20 Hz to 2048 Hz, covering the detected BNS inspiral[50]. For the EM signal, we use the data set compiled in Coughlin et al.[51], where in this work, we include the optical, infrared, and ultraviolet data between 0.5 and 10 days after the merger. The corresponding data is analyzed with a Gaussian Process Regression (GPR)-based kilonova model. For our analysis, we are presenting the best-fit lightcurve in Fig. 2, with its band representing a one

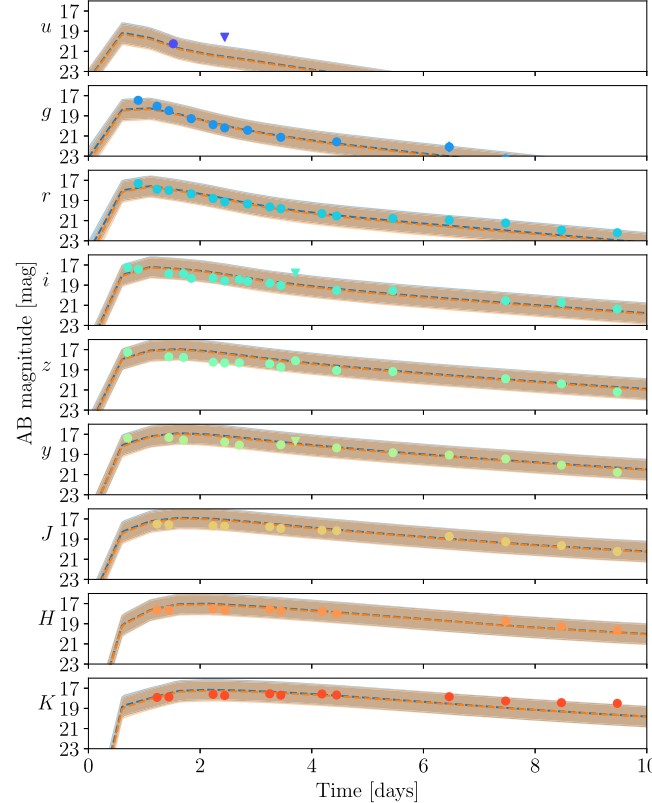

**Fig. 2 | Best-fit early-time lightcurve from the analysis.** The best-fit lightcurves (dashed, with the 1 magnitude uncertainty shown as the band) for AT2017gfo data when analyzing GW170817-and-AT2017gfo (orange) or GW170817-and-AT2017gfo-and-GRB170817A (blue) simultaneously. We note that both bands overlap almost completely, i.e., for AT2017gfo the accuracy of the kilonova lightcurve description does not depend noticeably on the inclusion of a GRB afterglow component. For the analysis, we restrict our dataset to times between 0.5 days up to 10 days after the BNS merger to simplify the joint GW170817-and-AT2017gfo-and-GRB170817A study as discussed in the main text.

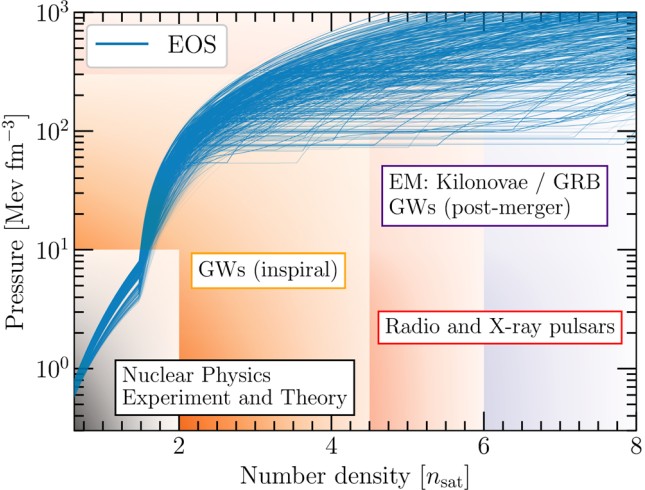

**Fig. 1 | Overview of constraints on the EOS from different information channels.** We show a set of possible EOSs (blue lines) that are constrained up to $1.5n_{sat}$ by Quantum Monte Carlo calculations using chiral EFT interactions[80] and extended to higher densities using a speed of sound model[149]. Different regions of the EOS can then be constrained by using different astrophysical messengers, indicated by rectangulars: GWs from inspirals of NS mergers, data from radio and X-ray pulsars, and EM signals associated with NS mergers. Note that the boundaries are not strict but depend on the EOS and properties of the studied system.

magnitude uncertainty for the individual lightcurves. This one magnitude uncertainty is introduced during the inference and should account for systematic uncertainties in kilonova modeling. In Supplementary information, we show how smaller or larger assumed uncertainties change our conclusions and it show that the one-magnitude is a sensible choice. Such a finding is also consistent with Heinzel et al.[52]. Therefore, we focus particularly on one-magnitude uncertainties' results. Nevertheless further work would be needed to understand in detail uncertainties related to the ejecta geometry[52], assumed heating rates, thermalization efficiencies and opacities within the ejecta[53]. Furthermore, we point out that for Fig. 2, we explicitly restricted our data set to the times between 0.5 to 10 days after the BNS merger, since model predictions at earlier or later times are more uncertain, e.g., due to less accurate opacities during early times and a larger impact of Monte Carlo noise in the employed radiative transfer models at late times. While this does not affect the GW170817-AT2017gfo analysis, it has an impact when we will also incorporate the GRB afterglow. In fact, we find that not restricting us to this time ranges can cause problems in the joint inference and it takes noticeably longer until the sampler converges.

Figure 3 summarizes our main findings and shows joint posteriors for the mass of the dynamical ejecta $m_{dyn}^{ej}$, the mass of the disk wind ejecta $m_{wind}^{ej}$, the chirp mass $\mathcal{M}_c$, the mass ratio $q$, the mass-weighted tidal deformability $\tilde{\Lambda}$, and the radius of a 1.4 solar mass neutron star $R_{1.4}$. In contrast to previous findings using simpler kilonova modeling (see ref. 54 and references therein), we can fit AT2017gfo with masses for the dynamical (about $0.006\,M_\odot$) and disk-wind (about $0.07\,M_\odot$) ejecta components that are within the range of values predicted by numerical-relativity simulations[55]. While the parameters extracted are consistent with our previous findings[22], we observe a clear improvement on the parameter error bounds due to (1) performing a simultaneous analysis of the distinct messengers and (2) employing a modified likelihood function when analyzing the kilonova. For instance, the constraints on $R_{1.4} = 11.86^{+0.41}_{-0.53}$, a typical choice to quantify EOS constraints, is significantly improved compared to our previous result, $R_{1.4} = 11.75^{+0.86}_{-0.81}$ km[22]. The half-width of $R_{1.4}$'s 90% credible interval decreases from about 800 m[22] to about 400 m.

### GW170817-AT2017gfo-GRB170817A

In addition to the combined analysis of GW170817 and AT2017gfo, we can also incorporate information obtained from the GRB afterglow of GW170817A, where we employ the data set collected in Troja et al.[56]. The GRB afterglow light-curve data are analyzed with the synthetic Gaussian jet-model lightcurve described before[57,58]. Figure 2 shows the corresponding best-fit lightcurve for the kilonova with a 1 magnitude uncertainty band as before. Moreover, we are also presenting the best-fit lightcurve, which includes kilonova and GRB afterglow, and the employed uncertainty band in Fig. 4. We find that both the kilonova AT2017gfo and the GRB afterglow GRB170817A are well described in our analysis.

Figure 3 again summarizes our findings for the joint posteriors of the mass of the dynamical ejecta, the mass of the disk wind ejecta, the on-axis isotropic equivalent energy, the chirp mass, the mass ratio, the mass-weighted tidal deformability, and the radius of a 1.4 solar mass neutron star for this analysis, which is consistent with GW170817-and-AT2017gfo only. Compared to the analysis of GW170817-and-AT2017gfo only, the improvement on the parameter uncertainties is minimal, yet, noticeable when information from GRB170817A is added. Although no significant constraint on the EOS is imposed by the jet energy $E_0$ as the ratio $\xi$ between it, $E_0$, and the disk mass $m_{disk}$ is taken as a free parameter, the inclination constraint from the GRB plays a role in the constraint on EOS. For an anisotropic kilonova model, the inclination angle changes the observable kilonova light curves beyond scaling (e.g., Fig. 2 in ref. 59), which is correlated with the ejecta masses

(e.g., Fig. 3 in ref. 60). Therefore the GRB's inclination measurement imposes a constraint on the EOS via the kilonova eject masses measurements.

Moreover, for future studies, we expect that the inclusion of the GRB afterglow will be of great importance for measuring the Hubble constant.

## Discussion

We have developed a publicly available NMMA framework for the interpretation and analysis of BNS and BHNS systems. This framework allows for the simultaneous analysis of GW and EM signals such as kilonovae and GRB afterglows. In addition, our framework allows us to incorporate constraints from nuclear-physics calculations, e.g., by sampling over EOS sets constrained by chiral EFT, and to include radio as well as X-ray measurements of isolated NSs. By employing our framework to a combined analysis of GW170817, AT2017gfo, and GRB170817A, we find that the radius of a typical 1.4 solar mass NS lies within $11.98^{+0.35}_{-0.40}$ km; cf. Table 1 for a selection of studies from the literature. Based on our findings, our analysis is a noticeable improvement over previous works. However, additional uncertainties in our work lie in limited physics input in kilonova and semi-analytic GRB and models. Therefore, reliable astrophysical interpretations of future BNS detections will only be possible if not only parameter estimation infrastructure, as presented in this work, but also the astrophysical models describing transient phenomena advance further. Nevertheless, given the increasing number of multi-messenger detections of BNS and BHNS merger, we expect to use our framework to further increase our knowledge about the interior of NSs during the coming years.

## Methods

### Equation of state construction

The EOS describes the relation between energy density $\varepsilon$, pressure $p$, and temperature $T$ of dense matter and additionally depends on the composition of the system. For NSs, thermal energies are much smaller than typical Fermi energies of the particles, and therefore, temperature effects can be neglected for isolated NSs or NSs in the inspiral phase of a merger. In these cases, the EOS simply relates $\varepsilon$ and $p$.

The most general constraints on the EOS can be inferred from the slope of the EOS, the speed of sound, defined as:

$$c_S = c\sqrt{\partial p/\partial \varepsilon}, \qquad (1)$$

where $c$ is the speed of light. Due to the laws of special relativity, the speed of sound has to be smaller than the speed of light, $c_S \le c$. Furthermore, the speed of sound in a NS has to be larger than zero, $c_S \ge 0$, as NSs would otherwise be unstable. These constraints alone, however, allow for an extremely large EOS space.

At nuclear densities, additional information on the EOS can be inferred from laboratory experiments and theoretical nuclear-physics calculations. For example, this information was used to constrain the properties of stellar matter in the NS crust[61,62], i.e., the outermost layer of NSs at densities below approximately $0.5n_{sat}$. Above roughly $0.5n_{sat}$, NS matter consists of a fluid of neutrons with a small admixture of protons. In this regime, the EOS can be constrained by microscopic calculations of dense nuclear matter. These calculations typically provide the energy per particle, $E/A(n,x)$, which is a function of density $n$ and proton fraction $x = n_p/n$ with $n_p$ being the proton density. From this, the EOS follows from:

$$\varepsilon(n,x) = n\frac{E}{A}(n,x), \qquad (2)$$

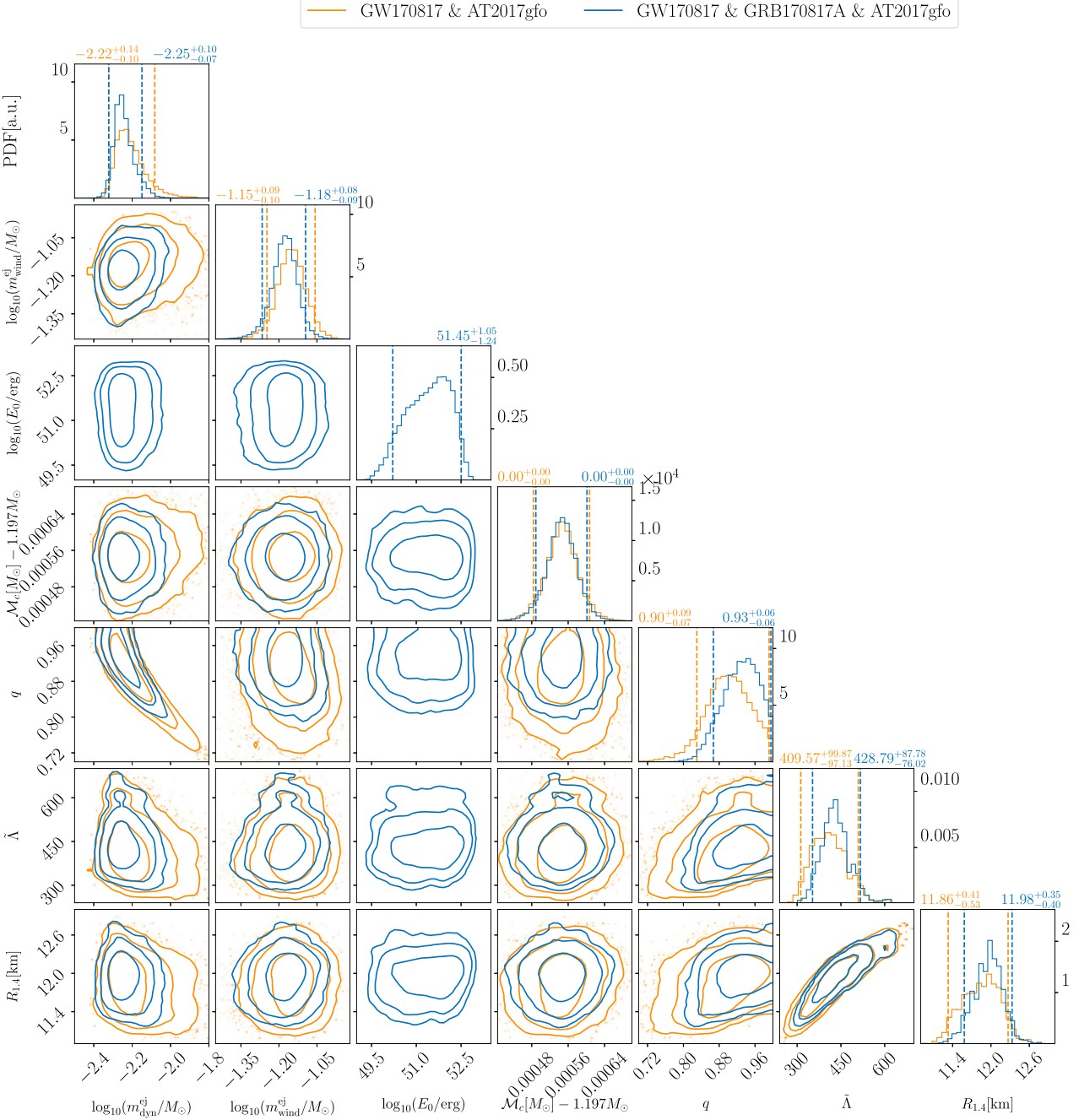

**Fig. 3 | Visualization of the posterior of the GW170817-and-AT2017gfo and GW170817-and-AT2017gfo-and-GRB170817A analysis.** Corner plot for the mass of the dynamical ejecta $m_{\mathrm{dyn}}^{\mathrm{ej}}$, the mass of the disk wind ejecta $m_{\mathrm{wind}}^{\mathrm{ej}}$, $\log_{10}$ of the GRB jet on-axis isotropic energy $\log_{10}E_0$, the detector-frame chirp mass $\mathcal{M}_c$, the mass ratio $q$, the mass-weighted tidal deformability $\tilde{\Lambda}$, and the radius of a 1.4 solar mass neutron star $R_{1.4}$ at 68%, 95% and 99% confidence. For the 1D posterior probability distributions, we mark the median (solid lines) and the 90% confidence interval (dashed lines) and report these above each panel. We show results that are based on the simultaneous analysis of GW170817-and-AT2017gfo (orange) and of GW170817-and-AT2017gfo-and-GRB170817A (blue).

and

$$p(n,x) = n^2 \frac{\partial E/A(n,x)}{\partial n}. \tag{3}$$

The proton fraction $x(n)$ is then determined from the beta equilibrium condition, $\mu_n = \mu_p + \mu_e$, where $\mu_i$ is the chemical potential of particle species $i$, and $n$, $p$, and $e$ refer to neutrons, protons, and electrons, respectively.

To calculate the energy per particle microscopically, one needs to solve the nuclear many-body problem, commonly described by the Schrödinger equation. This requires knowledge of the nuclear Hamiltonian describing the many-body system. Fundamentally, nuclear many-body systems are described by Quantum Chromodynamics (QCD), the fundamental theory of strong nuclear interactions. QCD describes the system in terms of the fundamental degrees of freedom (d.o.f.), quarks and gluons. Unfortunately, this approach is currently not feasible[63]. At densities of the order of $n_{\mathrm{sat}}$, however, the effective d.o.f. are nucleons, neutrons and protons, that can be treated as point-

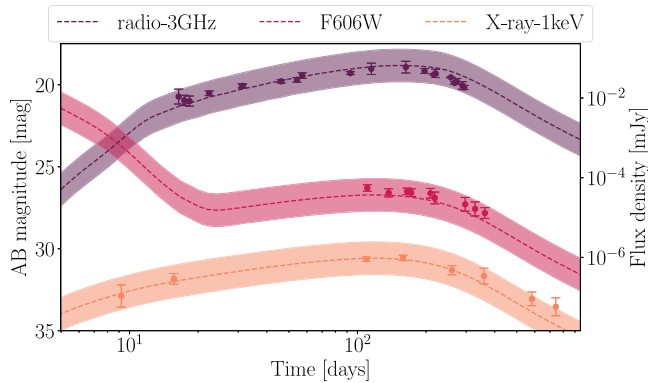

**Fig. 4 | Best-fit late-time lightcurve from the analysis.** The best-fit lightcurves (dashed, with the 1 magnitude uncertainty shown as the band) for the analysis of GRB170817A when simultaneously analyzing GW170817, AT2017gfo, GRB170817A. We compare our model predictions with the observational data including the 1-sigma measurement uncertainty.

**Table 1 | Comparison of radius measurements of a 1.4$M_\odot$ neutron star for a selection of multi-messenger studies**

| Reference | $R_{1.4}$ [km] |
|---|---|
| Dietrich et al.[22] | $11.75^{+0.86}_{-0.81}$ (90%) |
| Essick et al.[92] | $12.54^{+0.71}_{-0.63}$ (90%) |
| Breschi et al.[30] | $11.99^{+0.82}_{-0.85}$ (90%) |
| Nicholl et al.[31] | $11.06^{+1.01}_{-0.98}$ (90%) |
| Raaijmakers et al.[148] | $12.18^{+0.56}_{-0.79}$ (95%) |
| Miller et al.[96] | $12.45^{+0.65}_{-0.65}$ (68%) |
| Huth et al.[23] | $12.01^{+0.78}_{-0.77}$ (90%) |
| this work [NMMA] | $11.98^{+0.35}_{-0.40}$ (90%) |

A selected list of radius measurements of a 1.4$M_\odot$ neutron star from various multi-messenger studies is shown. We denote the corresponding credible interval in parenthesis.

like nonrelativistic particles. Then, the nuclear Hamiltonian can be written generically as:

$$H = T + \sum_{i<j} V_{ij}^{NN} + \sum_{i<j<k} V_{ijk}^{3N} + \cdots, \qquad (4)$$

where $T$ denotes the kinetic energy of the nucleons, $V_{ij}^{NN}$ describes two-nucleon (NN) interactions between nucleons $i$ and $j$, and $V_{ijk}^{3N}$ describes three-nucleon (3N) interactions between nucleons $i$, $j$, and $k$. In principle, interactions involving four or more nucleons can be included, but initial studies have found these to be small compared to present uncertainties[64].

The derivation of the nuclear Hamiltonian (Eq. (4)) from QCD is not feasible due to its nonperturbative nature. In this work, we therefore use a common approach and choose nucleons as effective d.o.f. The interactions among nucleons can then be derived in the framework of Chiral effective field theory (EFT)[65,66]. Chiral EFT starts out with the most general Lagrangian consistent with all the symmetries of QCD in terms of nucleonic degrees of freedom. It explicitly includes meson-exchange interactions for the lightest mesons, i.e., the pions. This approach yield an infinite number of pion-exchange and nucleon-contact interactions which needs to be organized in terms of a hierarchical expansion in powers of a soft (low-energy) scale over a hard (high-energy) scale. In chiral EFT, the soft scale $q$ is given by the nucleons' external momenta or the pion mass. The hard scale, also called the breakdown scale $\Lambda_b$, is of the order of 500–600 MeV[67] and interaction contributions involving heavier d.o.f., such as the $\rho$ meson, are integrated out. The chiral Lagrangian is then expanded in powers

of $q/\Lambda_b$ according to a power-counting scheme. Most current chiral EFT interactions are derived in Weinberg power counting[65,66,68–70]. One can then derive the nuclear Hamiltonian from this chiral Lagrangian in a consistent order-by-order framework that allows for an estimate of the theoretical uncertainties[67,71,72] and that can be systematically improved by increasing the order of the calculation. Chiral EFT Hamiltonian naturally include NN, 3N, and higher many-body forces, see Eq. (4), and chiral EFT predicts a natural hierarchy of these contributions. For example, 3N interactions start to contribute at third order (N²LO) in the expansion. Typical state-of-the art calculations truncate the chiral expansion at N²LO[37,40,73] or fourth order (N³LO)[39,74].

With the nuclear Hamiltonian at hand, one then needs to solve the many-body Schrödinger equation which requires advanced numerical methods. Examples of such many-body techniques include many-body perturbation theory (MBPT)[38,39,74], the self-consistent Green's function (SCGF) method[75], or the coupled-cluster (CC) method[73,76]. Here, we employ Quantum Monte Carlo (QMC) methods[77], which provide non-perturbative solutions of the Schrödinger equation. QMC methods are stochastic techniques which treat the Schrödinger equation as a dif-fusion equation in imaginary time. In the QMC framework, one begins by choosing a trial wavefunction of the many-body system, which for nuclear matter can be described as a slater determinant of non-interacting fermions multiplied with NN and 3N correlation functions. This trial wavefunction is evolved to large imaginary times, projecting out high-energy excitations, and converging to the true ground state of the system as long as the trial wavefunction has a non-zero overlap with it. Among QMC methods, two well-established algorithms are Green's function Monte Carlo (GFMC), used to describe light atomic nuclei with great precision[77], and Auxiliary Field Diffusion Monte Carlo (AFDMC)[78], suitable to study larger systems such as nuclear matter. Here, we employ AFDMC calculations of neutron matter but our NMMA framework is sufficiently flexible to employ any low-density calculation for neutron-star matter. We then extend our neutron-matter calculations to neutron-star conditions by extrapolating the calculations to $\beta$ equilibrium using phenomenological information on symmetric nuclear matter and constructing a consistent crust reflecting the uncertainties of the calculations[79]. This crust includes a description of the outer crust[61] and uses the Wigner-Seitz approx-imation to calculate the inner-crust EOS consistently with our AFDMC calculations.

At nuclear densities, chiral EFT together with a suitable many-body framework provides for a reliable description of nuclear matter with systematic uncertainty estimates. With increasing density, however, the associated theoretical uncertainty grows fast due to the correspondingly larger nucleon momenta approaching the break-down scale. The density up to which chiral EFT remains valid is not exactly known but estimates place it around $2n_{sat}$[67,80]. Hence, chiral EFT calculations constrain the EOS only up to these densities but to explore the large EOS space beyond the breakdown of chiral EFT, one requires a physics-agnostic extension scheme. Here, physics-agnostic implies that no model assumptions, e.g., about the exis-tence of certain d.o.f. at high densities, are made. Instead, the EOS is only bounded by conditions of causality, $c_S \leq c$, and mechanical sta-bility, $c_S \geq 0$, mentioned before. There exist several such extension schemes in literature: parametric ones, like the polytropic expansion[81–83] or expansions in the speed of sound[84,85], and non-parametric approaches[86]. To extend the AFDMC calculations employed here, we employ a parametric speed-of-sound extension scheme. Working in the $c_S$ versus $n$ plane, the speed of sound $c_S(n)$ is determined with theoretical uncertainty estimates by chiral EFT up to a reference density below the expected breakdown density. From this uncertainty band, we sample a speed-of-sound curve up to the reference density. Beyond this density, we create a typically non-uniform grid in density up to a large density $\approx 12n_{sat}$, well beyond the regime realized in NSs. For each grid point, we sample random values

for $c_s^2(n_i)$ between 0 and $c^2$ (we set $c = 1$ in the following). We then connect the chiral EFT draw for the speed of sound with all points $c_{s,i}^2(n_i)$ using linear segments. The resulting density-dependent speed of sound can be integrated to give the EOS, i.e., the pressure, baryon density, and energy density. In the interval $n_i \leq n \leq n_{i+1}$:

$$p(n) = p(n_i) + \int_{n_i}^{n} c_s^2(n')\mu(n')dn', \tag{5}$$

$$\epsilon(n) = \epsilon(n_i) + \int_{n_i}^{n} \mu(n')dn', \tag{6}$$

where $\mu(n)$ is the chemical potential that can be obtained from the speed of sound using the relation:

$$\mu(n) = \mu_i \exp\left[\int_{\log n_i}^{\log n} c_s^2(\log n')d\log n'\right]. \tag{7}$$

For each reconstructed EOS, constrained by Chiral EFT at low densities and extrapolated via the $c_S$ extension to larger densities, the global properties of NSs can be calculated by solving the Tolman-Oppenheimer-Volkoff (TOV) equations. This way, we determine the NS radii ($R$) and dimensionless tidal deformabilities ($\Lambda$) as functions of their masses ($M$). We repeat this approach for a large number of samples to construct EOS priors for further analyses of NS data.

This approach is flexible and additional information on high-density phases of QCD can be included straightforwardly. For example, pQCD calculations at asymptotically high densities[47], of the order of $40$–$50 n_{sat}$, might be used to constrain the general EOS extension schemes even further[48,83]. However, the exact impact of these constraints at densities well beyond the regime realized in NSs needs to be studied in more detail. While our NMMA framework currently does not have this capability, we are planning to add this in the near future. Similarly, instead of using general extension models, one can employ specific high-density models accounting for quark and gluon d.o.f. One such model is the quarkyonic-matter model[87–90], which describes the observed behavior of the speed of sound in NSs[80]: a rise of the speed of sound at low densities to values above the conformal limit of $c/\sqrt{3}$, followed by a decrease to values below the conformal limit at higher densities. In future work, we will address quarkyonic matter and other models in our NMMA framework.

The construction of the EOS, as detailed above, is implemented in the NMMA code under the class `EOS_with_CSE`. This class allows for (1) an exploration of theoretical uncertainties in the low-density EOS and (2) constructs the high-density EOS using a $c_S$ extrapolation. (1) Low-density uncertainties are implemented by requiring two tabulated EOS files for the lower and upper bound of the uncertainty band as inputs, containing the pressure, energy density and number density up to the chosen breakdown density of the model. By default, the results of a QMC calculation using local chiral EFT interactions at N²LO[80] with theoretical uncertainties are provided. Upon initiation of the class, a sample is drawn from the low-density uncertainty band using a 1-parameter sampling technique. In this approach, a uniform random number $\omega$ is sampled uniformly between 0 and 1, and the interpolated EOS is given as:

$$p(n) = p_{soft}(n) + \omega(p_{stiff}(n) - p_{soft}(n)), \tag{8}$$

$$\epsilon(n) = \epsilon_{soft}(n) + \omega(\epsilon_{stiff}(n) - \epsilon_{soft}(n)), \tag{9}$$

where the subscripts "soft" and "stiff" refer to the lower and upper bounds of the EFT uncertainty band, respectively. This sampling technique assumes that pressure and energy density are correlated but

we have found that releasing this assumption and using a four-parameter form suggested by Gandolfi et al.[91] does not change our results appreciably. In future, we will explore additional schemes, e.g., using Gaussian processes[92].

(2) The EOS given by Eqs. (8) and (9) is used up to a breakdown density determined by the user. By default, this density is set to $2n_{sat}$. Beyond this density, the class constructs the EOS using a $c_S$ extension. The maximum density up to which the EOS is extrapolated and the number of linear line segments can be adjusted by the user, with the default values being $12n_{sat}$ for the former and 5 line segments for the latter. The code then solves Eqs. (5)–(7) to give the extrapolated EOS. The pressure, energy density, and number density describing the full EOS are accessible as attributes of the `EOS_with_CSE` class.

Finally, the method `construct_family` solves the stellar structure equations (TOV equations and equations for the quadrupole perturbation of spherical models), and returns a sequence of NSs with their masses, radii and dimensionless tidal deformabilities as arrays.

### Prior weighting to incorporate radio and X-ray observations of single neutron stars

To incorporate mass measurements of heavy pulsars and mass-radius measurements of isolated pulsars, the associated likelihood is calculated and taken as the prior probability for an EOS for further analysis. For instance, the radio observations on PSR J0348+4042[44], and PSR J1614-2230[45] provide a lower bound on the maximum mass of a NS.

The likelihood for a mass-only measurement is given by:

$$\mathcal{L}_{PSR-mass}(\mathbf{E}) = \int_0^{M_{TOV}} dM\, \mathcal{P}(M|PSR), \tag{10}$$

where $\mathcal{P}(M|PSR)$ is the posterior distribution of the pulsar's mass and $M_{TOV}$ is the maximum mass supported by the EOS with parameters $\mathbf{E}$. The posterior distributions of pulsar masses are typically well approximated by Gaussians[22].

Recent X-ray observations of millisecond pulsars by NASA's Neutron Star Interior Composition Explorer (NICER) mission have been used to simultaneously determine the mass and radius of these NSs[93–97]. The corresponding likelihood is given by:

$$\begin{aligned}\mathcal{L}_{NICER}(\mathbf{E}) &= \int dM \int dR\, \mathcal{P}_{NICER}(M,R)\frac{\pi(M,R|\mathbf{E})}{\pi(M,R|I)} \\ &\propto \int dM \int dR\, \mathcal{P}_{NICER}(M,R)\delta(R - R(M;\mathbf{E})) \\ &\propto \int dM\, \mathcal{P}_{NICER}(M,R = R(M;\mathbf{E})),\end{aligned} \tag{11}$$

where $\mathcal{P}_{NICER}(M,R)$ is the joint-posterior distribution of mass and radius as measured by NICER and we use the fact that (1) the radius is a function of mass for a given EOS, and (2) that without further EOS information, e.g., through chiral EFT, the prior for the radius given mass is taken to be uniform.

### Gravitational-wave inference
**GW models.** A complex frequency-domain GW signal is given by:

$$h(f) = A(f)e^{-i\psi(f)}, \tag{12}$$

with the amplitude $A(f)$ and the GW phase $\psi(f)$. Because of the NS's finite size and internal structure, BNS and BHNS waveform models have to incorporate tidal contributions for an accurate interpretation of the binary coalescence. Such tidal contributions account for the deformation of the stars in their companions' external gravitational field[98,99] and, once measured, allow to place constraints on the EOS governing the NS interior[100–103]. They are attractive because they convert energy from the orbital motion to a deformation of the stars, and

lead to an accelerated inspiral. In the case of non-spinning compact objects, the leading-order tidal contribution depends on the tidal deformability:

$$\tilde{\Lambda} = \frac{16}{13} \frac{(m_1 + 12 m_2) m_1^4 \Lambda_1 + (m_2 + 12 m_1) m_2^4 \Lambda_2}{(m_1 + m_2)^5} \tag{13}$$

with the individual tidal deformabilities $\Lambda_{1,2} = \frac{2}{3} k_2^{1,2} / C_{1,2}^5$ and the individual masses $m_{1,2}$. Here, $k_2^{1,2}$ are the Love numbers describing the static quadrupole deformation of one body inside the gravitoelectric field of the companion and $C_{1,2}$ are the individual compactnesses $C_{1,2} = m_{1,2}/R_{1,2}$ in isolation.

To date, there are three different types of BNS or BHNS models for the inspiral GW signal that are commonly used: Post-Newtonian (PN) models[104–107], effective-one-body (EOB) models[108–116], and phenomenological approximants[117–121]. In the NMMA framework, we make use of the LALSuite[122] software package, in particular LALSimulation, so that the BNS and BHNS models used by the LIGO-Virgo-Kagra Collaborations can be easily employed. This includes:

- PN models such as `TaylorT2`, `TaylorT4`, or `TaylorF2` where a PN descriptions for the point-particle BBH baseline as well as the tidal description is employed.
- the most commonly used tidal EOB models `SEOBNRv4T`[111,116,123], its frequency-domain surrogate model[124], as well as the `TEO-BResumS` model[112,125] including its post-adiabatic accelerated version[126] which enables it being used during parameter estimation.
- and phenomenological models such as `IMRPhenomD_NRTidal`, `SEOBNRv4_ROM_NRTidal`, `IMRPhenomPv2_NRTidal`, `IMRPhenomD_NRTidalv2`, `SEOBNRv4_ROM_NRTidalv2`, `IMRPhenomPv2_NRTidalv2`[118–120], `PhenomNSBH`, and `SEOBNRv4_ROM_NRTidalv2_NSBH`[121,127].

**GW analysis.** By assuming stationary Gaussian noise, the GW likelihood $\mathcal{L}_{GW}(\boldsymbol{\theta})$ that the data $d$ is a sum of noise and a GW signal $h$ with parameters $\boldsymbol{\theta}$ is given by[128]:

$$\mathcal{L}_{GW} \propto \exp\left( -\frac{1}{2} \langle d - h(\boldsymbol{\theta}) | d - h(\boldsymbol{\theta}) \rangle \right), \tag{14}$$

where the inner product $\langle a | b \rangle$ is defined as:

$$\langle a | b \rangle = 4 \Re \int_{f_{low}}^{f_{high}} \frac{\tilde{a}(f) \tilde{b}^*(f)}{S_n(f)} df. \tag{15}$$

Here, $\tilde{a}(f)$ is the Fourier transform of $a(t)$, $^*$ denotes complex conjugation, and $S_n(f)$ is the one-sided power spectral density of the noise. The choice of $f_{low}$ and $f_{high}$ depends on the type of binary that we are interested in. In our study, we will set $f_{low}$ and $f_{high}$ to 20 Hz and 2048 Hz, respectively. This is sufficient for capturing the inspiral up to the moment of merger for a typical BNS system in the advanced GW detector era.

### Electromagnetic signals

**Kilonova models.** Kilonova models are extracted using the 3D Monte Carlo radiative transfer code POSSIS[129]. The code can handle arbitrary geometries for the ejected material and produces spectra, lightcurves and polarization as a function of the observer viewing angle. Given an input model with defined densities $\rho$ and compositions (i.e., electron fraction $Y_e$), the code generates Monte Carlo photon packets with initial location and energy sampled from the energy distribution from radioactive decay of r-process nuclei within the model. The latter depends on the mass/density distribution of the model and the assumed nuclear heating rates and thermalization efficiencies. The frequency of each Monte Carlo photon packet is sampled according to the temperature $T$ in the ejecta, which is calculated at each time-step[130,131]. Photon packets

are then followed as they diffuse out of the ejected material and interact with matter via either electron scattering or bound-bound line transitions. Time- and wavelength-dependent opacities $\kappa_\lambda(\rho, T, Y_e, t)$ from Tanaka et al.[132] are implemented in the code and depend on the local properties of the ejecta ($\rho$, $T$, and $Y_e$). Spectral time series are extracted using the technique described by Bulla et al.[133] and used to construct broad-band lightcurves in any desired filter.

**Supernova models.** Templates available within the `SNCosmo` library[134] are used to model supernova spectra. Currently, the `salt2` model for Type Ia supernovae and the `nugent-hyper` model for hypernovae associated with long GRBs are implemented in the framework and have been used in the past[36]. However, the framework is flexible enough such that additional templates for different types of supernovae can be added with minimal effort.

**Kilonova/supernova inference.** Our EM inference of kilonovae and GRB afterglows is based on the AB magnitude for a specific filter $j, m_i^j(t_i)$. We assume these measurements to be given as a time series at times $t_i$ with a corresponding statistical error $\sigma_i^j \equiv \sigma^j(t_i)$. The likelihood function $\mathcal{L}_{EM}(\boldsymbol{\theta})$ then reads[135]:

$$\mathcal{L}_{EM} \propto \exp\left( -\frac{1}{2} \sum_{ij} \frac{(m_i^j - m_i^{j,est}(\boldsymbol{\theta}))^2}{(\sigma_i^j)^2 + \sigma_{Sys}^2} \right), \tag{16}$$

where $m_i^{j,est}(\boldsymbol{\theta})$ is the estimated AB magnitude for the parameters $\boldsymbol{\theta}$ and $\sigma_{sys}$ is the additional error budget for accounting the systematic uncertainty within the electromagnetic signal modeling. The inclusion of $\sigma_{sys}$ is equivalent to adding a shift of $\Delta m$ to the light curve, for which marginalized with respect to a zero-mean normal distribution with a variance of $\sigma_{sys}^2$.

This likelihood is equivalent to approximating the probability distribution of the spectral flux density $f_\nu$ to be a Log-normal distribution. The Log-normal distribution is a 2-parameter maximum entropy distribution with its support equals to the possible range for $f_\nu \in (0, \infty)$. There are two advantages of approximating $f_\nu$ with a Log-normal distribution: (1) if the uncertainty is larger or comparable to the measured value, it avoids having non-zero support for the nonphysical $f_\nu < 0$; (2) if the uncertainty is much smaller than the measured value, the Log-normal distribution approaches the normal distribution.

For kilonovae, we use the same model presented in Dietrich et al.[22]. The model is controlled by four parameters, namely, the dynamical ejecta mass $m_{dyn}^{ej}$, the disk wind ejecta mass $m_{wind}^{ej}$, the half-opening angle of the lanthanide-rich component $\Phi$, and the viewing angle $\theta_{obs}$.

**GRB afterglows.** In our framework, the computation of the GRB afterglow lightcurves is until now based on the publicly available semi-analytic code `afterglowpy`[57,58]. The inclusion of other afterglow models is currently ongoing.

The GRB afterglow emission is produced by relativistic electrons gyrating around the magnetic field lines. These electrons are accelerated by the Fermi first-order acceleration (diffusive shock acceleration) and the magnetic field is assumed to be of turbulent nature, amplified by processes acting in collision-less shocks. The complex physics of electron acceleration at shocks is approximated by the equipartition parameters, $\epsilon_e$ and $\epsilon_B$, denoting the fraction of the shock energy that goes into the relativistic electrons and magnetic field, respectively, and $p$, and the slope of the electron energy distribution $dn/d\gamma \propto \gamma^{-p}$, with $n$ being the electron number density and $\gamma$ being the electron Lorentz factor. The flux density of the curvature radiation is:

$$F_\nu = \frac{1}{4\pi d_L^2} \int d\theta d\phi R^2 \sin(\theta) \frac{\epsilon_\nu}{\alpha_\nu} (1 - e^{-\tau}), \tag{17}$$

where $\tau$ is the optical depth and $\epsilon_\nu$ and $\alpha_\nu$ are the impassivity coefficient and absorption coefficient, respectively. For a fixed power-law distribution of electrons these can be approximated analytically[136]. The synchrotron self-absorption is neglected in this work.

In order to capture the possible dependence of the GRB properties on the polar angle, the jet is discretized into a set of lateral axisymmetric (conical) layers, each of which is characterized by its initial velocity, mass, and angle. Several prescriptions for the initial angular distribution of the jet energy are available in the code. As default, we use the Gaussian jet model with $E \propto E_0 \exp(-\frac{1}{2}(\frac{\theta}{\theta_c})^2)$, where $\theta_c$ characterizes the width of the Gaussian. The jet truncation angle is $\theta_w$. We assume the GRB jet to be powered by the accretion of mass from the disk onto the remnant black hole[137–140]. Consequently, the jet energy is proportional to the leftover disk mass:

$$E_0 = \epsilon \times (1 - \xi) \times m_{disk}, \qquad (18)$$

where $\xi$ is the fraction of disk mass ejected as wind and $\epsilon$ is the fraction of residual disk mass converted into jet energy.

The dynamical evolution of these layers is computed semi-analytically using the "thin-shell approximation" casting energy-conservation equations and shock-jump conditions into a set of evolution equations for the blast wave velocity and radius. Within blast waves, the pressure gradient perpendicular to the normal leads to lateral expansion[141,142]. In other words, the transverse pressure gradient adds the velocity along the tangent to the blast wave surface, forcing the latter to expand. The lateral expansion is important for late-time afterglow and is included in the code.

Finally, the flux density, $F_\nu$, is obtained by equal arrival time surface integration, Eq. (17), taking into account relativistic effects, i.e., that the observed $F_\nu$ is composed of contributions from different blast waves that has emitted at different comoving time and at different frequencies.

**Connecting electromagnetic signals to source properties.** To connect the observed GRB, kilonova, and GRB afterglow properties to the binary properties, we rely on phenomenological relations, i.e., fits based on numerical-relativity simulations. For our work, we use the fits presented in Kruger et al.[143] and Dietrich et al.[22] but emphasize that a variety of other fitting formulas exist in the literature[20,51,55,144,145].

In NMMA, the dynamical ejecta mass $m_{dyn}^{ej}$ is connected to the binary properties through the phenomenological relation[143]:

$$\frac{m_{dyn,fit}^{ej}}{10^{-3} M_\odot} = \left( \frac{a}{C_1} + b \left( \frac{m_2}{m_1} \right)^n + c C_1 \right) + (1 \leftrightarrow 2), \qquad (19)$$

where $m_i$ and $C_i$ are the masses and the compactness of the two components of the binary with best-fit coefficients $a = -9.3335$, $b = 114.17$, $c = -337.56$, and $n = 1.5465$. This relation enables an accurate estimation of the ejecta mass with an error well-approximated by a zero-mean Gaussian with a standard deviation $0.004 M_\odot$[143]. Therefore, the dynamical ejecta mass can be approximated as:

$$m_{dyn}^{ej} = m_{dyn,fit}^{ej} + \alpha, \qquad (20)$$

where $\alpha \sim \mathcal{N}(\mu = 0, \sigma = 0.004 M_\odot)$.

To determine the disk mass $m_{disk}$, we follow the description of Dietrich et al.[22]:

$$\log_{10}\left( \frac{m_{disk}}{M_\odot} \right) = \max\left( -3, a \left( 1 + b \tanh\left( \frac{c - (m_1 + m_2) M_{threshold}^{-1}}{d} \right) \right) \right), \qquad (21)$$

with $a$ and $b$ given by:

$$a = a_o + \delta a \cdot \Delta, \qquad b = b_o + \delta b \cdot \Delta, \qquad (22)$$

where $a_o, b_o, \delta a, \delta b, c,$ and $d$ are free parameters. The parameter $\Delta$ is given by:

$$\Delta = \frac{1}{2} \tanh\left( \beta(q - q_{trans}) \right), \qquad (23)$$

where $q \equiv m_2/m_1 \leq 1$ is the mass ratio and $\beta$ and $q_{trans}$ are free parameters. The best-fit model parameters are $a_o = -1.581, \delta a = -2.439, b_o = -0.538, \delta b = -0.406, c = 0.953, d = 0.0417, \beta = 3.910, q_{trans} = 0.900$. The threshold mass $M_{threshold}$ for a given EOS is estimated as[146]:

$$M_{threshold} = \left( 2.38 - 3.606 \frac{M_{TOV}}{R_{1.6}} \right) M_{TOV}, \qquad (24)$$

where $M_{TOV}$ and $R_{1.6}$ are the maximum mass of a non-spinning NS and the radius of a $1.6 M_\odot$ NS. We note that we assume that the disk-wind ejecta component is proportional to the disk mass, i.e., $m_{wind}^{ej} = \xi \times m_{disk}$.

**Bayesian statistics.** Based on Bayes' theorem, the posterior distribution of the parameters $p(\boldsymbol{\theta}|d,\mathcal{H})$ under hypothesis $\mathcal{H}$ with data $d$ is given by:

$$p(\boldsymbol{\theta}|d,\mathcal{H}) = \frac{p(d|\boldsymbol{\theta},\mathcal{H})p(\boldsymbol{\theta}|\mathcal{H})}{p(d|\mathcal{H})} \equiv \frac{\mathcal{L}(\boldsymbol{\theta})\pi(\boldsymbol{\theta})}{\mathcal{Z}(d)}, \qquad (25)$$

where $\mathcal{L}(\boldsymbol{\theta}), \pi(\boldsymbol{\theta})$, and $\mathcal{Z}(d)$ are the likelihood, prior, and evidence, respectively. The prior describes our knowledge of the source or model parameters prior to the experiment or observation. The likelihood and evidence quantify how well the hypothesis describes the data for a given set of parameters and over the whole parameter space, respectively. Throughout our NMMA pipeline, all data analyses use Bayes' theorem but differences appear due to the functional form of the likelihood and its specific dependence on the source parameters. For example, the GW likelihood is evaluated with a cross-correlation between the data and the GW waveform and the EM signal analysis employs a $\chi^2$ log-likelihood between the predicted lightcurves with the observed apparent magnitude data, however, from a Bayesian viewpoint their treatment is equivalent only with different likelihood functions.

In addition to the posterior estimation, the evidence $\mathcal{Z}$ carries additional information on the plausibility of a given hypothesis $\mathcal{H}$. The evidence is given by:

$$\mathcal{Z}(d|\mathcal{H}) = \int d\boldsymbol{\theta} p(d|\boldsymbol{\theta},\mathcal{H})p(\boldsymbol{\theta}|\mathcal{H}) = \int d\boldsymbol{\theta} \mathcal{L}(\boldsymbol{\theta})\pi(\boldsymbol{\theta}), \qquad (26)$$

which is the normalization constant for the posterior distribution. Moreover, we can compare the plausibilities of two hypotheses, $\mathcal{H}_1$ and $\mathcal{H}_2$, by using the odd ratio $\mathcal{O}_2^1$, which is given by:

$$\mathcal{O}_2^1 = \frac{\mathcal{Z}_1}{\mathcal{Z}_2} \frac{p(\mathcal{H}_1)}{p(\mathcal{H}_2)} \equiv \mathcal{B}_2^1 \Pi_2^1, \qquad (27)$$

where $\mathcal{B}_2^1$ and $\Pi_2^1$ are the Bayes factor and prior odds, respectively. If $\mathcal{O}_2^1 > 1, \mathcal{H}_1$ is more plausible than $\mathcal{H}_2$, and vice versa.

## Data availability
The datasets generated during the current study are available in the Zenodo repository https://doi.org/10.5281/zenodo.6551053. The GW

data strain that we have analyzed in this work was obtained from the Gravitational Wave Open Science Center (ref. 147 at https://www.gw-openscience.org), and the NICER data were obtained from Zenodo (10.5281/zenodo.3473466, 10.5281/zenodo.4670689 and 10.5281/zenodo.4697625). Source data are provided with this paper.

## Code availability

The source code of the NMMA framework, which was used for this study, is publicly available at https://github.com/nuclear-multimessenger-astronomy/nmma. In addition, all employed GW models are available on https://git.ligo.org/lscsoft. The bilby and parallel bilby software packages are available at https://git.ligo.org/lscsoft/bilby and https://git.ligo.org/lscsoft/parallel_bilby, respectively.

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

## Acknowledgements

We thank N. Andersson, R. Essick, P. Landry, and J. Margueron for insightful discussions. P.T.H.P. and C.V.D.B. are supported by the research program of the Netherlands Organization for Scientific Research (NWO). T.D. acknowledges support of the Daimler and Benz Foundation. M.W.C. acknowledges support from the National Science Foundation with grant numbers PHY-2308862 and OAC-2117997. M.B. acknowledges support from the Swedish Research Council (Reg. no. 2020-03330). The work of I.T. was supported by the U.S. Department of Energy, Office of Science, Office of Nuclear Physics, under contract No. DE-AC52-06NA25396, by the Laboratory Directed Research and Development program of Los Alamos National Laboratory under project number 20220658ER, and by the U.S. Department of Energy, Office of Science, Office of Advanced Scientific Computing Research, Scientific Discovery through Advanced Computing (SciDAC) program. J.H. acknowledges support from the National Science Foundation with grant number PHY-1806990. Funded/Co-funded by the European Union (ERC, SMArt, 101076369). Views and opinions expressed are however those of the author(s) only and do not necessarily reflect those of the European Union or the European Research Council. Neither the European Union nor the granting authority can be held responsible for them. Computations have been performed on the Minerva HPC cluster of the Max-Planck-Institute for Gravitational Physics and on SuperMUC-NG (LRZ) under project number pn56zo. Computational resources have also been provided by the Los Alamos National Laboratory Institutional Computing Program, which is supported by the U.S. Department of Energy National Nuclear Security Administration under Contract No. 89233218CNA000001, and by the National Energy Research Scientific Computing Center (NERSC), which is supported by the U.S. Department of Energy, Office of Science, under contract No. DE-AC02-05CH11231. Resources supporting this work were provided by the Minnesota Supercomputing Institute (MSI) at University of Minnesota under the project "Identification of Variable Objects in the Zwicky Transient Facility," and the Supercomputing Laboratory at King Abdullah University of Science and Technology (KAUST) in Thuwal, Saudi Arabia. This research has made use of data, software and/or web tools obtained from the Gravitational Wave Open Science Center (https://www.gw-openscience.org), a service of LIGO Laboratory, the LIGO Scientific Collaboration and the Virgo Collaboration. This material is based upon work supported by NSF's LIGO Laboratory which is a major facility fully funded by the National Science Foundation. Virgo is funded by the French Centre National de Recherche Scientifique (CNRS), the Italian Istituto Nazionale della Fisica Nucleare (INFN) and the Dutch Nikhef, with contributions by Polish and Hungarian institutes.

## Author contributions

Conceptualization: P.T.H.P., T.D., M.W.C., M.B., I.T., and C.V.D.B.; Methodology: P.T.H.P., T.D., M.W.C., M.B., I.T., M.A., S.A., and C.V.D.B.; Data curation: P.T.H.P., M.W.C., and M.A.; Software: P.T.H.P., T.D., M.W.C., M.B., I.T., M.A., T.B., R.W.K., N.K., G.M., B.R., N.S., A.T., S.A., R.O.V., J.H., P.S., R.Sh., R.So., and C.V.D.B.; Validation: P.T.H.P. and M.W.C.; Formal analysis: P.T.H.P. and M.W.C.; Resources: T.D., M.W.C., M.B., and I.T.; Funding acquisition: T.D., M.W.C., M.B., and I.T.; Project administration: P.T.H.P., T.D., M.W.C., M.B., I.T., and C.V.D.B.; Supervision: P.T.H.P., T.D., M.W.C., M.B., and I.T.; Visualization: P.T.H.P., T.D., M.W.C., M.B., I.T., and N.K.; Writing—original draft: P.T.H.P., T.D., M.W.C., M.B., I.T., N.K., V.N., and P.S.; Writing—review and editing: P.T.H.P., T.D., M.W.C., M.B., I.T., N.K., and V.N.

## Funding

## Competing interests

The authors declare no competing interests.

## Additional information

¹Nikhef, Science Park 105, 1098 XG Amsterdam, The Netherlands. ²Institute for Gravitational and Subatomic Physics (GRASP), Utrecht University, Princetonplein 1, 3584 CC Utrecht, The Netherlands. ³Institut für Physik und Astronomie, Universität Potsdam, Haus 28, Karl-Liebknecht-Str. 24/25, 14476 Potsdam, Germany. ⁴Max Planck Institute for Gravitational Physics (Albert Einstein Institute), Am Mühlenberg 1, 14476 Potsdam, Germany. ⁵School of Physics and Astronomy, University of Minnesota, Minneapolis, MN 55455, USA. ⁶The Oskar Klein Centre, Department of Astronomy, Stockholm University, AlbaNova, SE-106 91 Stockholm, Sweden. ⁷Department of Physics and Earth Science, University of Ferrara, Via Saragat 1, I-44122 Ferrara, Italy. ⁸INFN, Sezione di Ferrara, Via Saragat 1, I-44122 Ferrara, Italy. ⁹INAF, Osservatorio Astronomico d'Abruzzo, Via Mentore Maggini snc, 64100 Teramo, Italy. ¹⁰Theoretical Division, Los Alamos National Laboratory, Los Alamos, NM 87545, USA. ¹¹Department of Physics, American University of Sharjah, PO Box 26666 Sharjah, UAE. ¹²Laboratoire de Physique et de Chimie de l'Environnement, Université Joseph KI-ZERBO, Ouagadougou, Burkina Faso. ¹³Observatoire de la Côte d'Azur, Université Côte d'Azur, CNRS, 96 Boulevard de l'Observatoire, F06304 Nice Cedex 4, France. ¹⁴Department of Physics, American University, Washington, DC 20016, USA. ¹⁵Department of Physics and Astronomy, University of Minnesota—Duluth, Duluth, MN 55812, USA. ¹⁶Department of Physics, Drexel University, Philadelphia, PA 19104, USA. ¹⁷Department of Physics, Massachusetts Institute of Technology, 77 Massachusetts Ave, Cambridge, MA 02139, USA. ¹⁸Department of Physics, Deshbandhu College, University of Delhi, New Delhi, India. ¹⁹Université Lyon, Université Claude Bernard Lyon 1, CNRS/IN2P3, IP2I Lyon, UMR 5822, F-69622 Villeurbanne, France. ²⁰Department of Physics, Syracuse University, Syracuse, NY 13244, USA. ✉e-mail: tim.dietrich@uni-potsdam.de

