## [Peer Review File · Nature Communications]

nature portfolio

Peer Review File

Editorial Note: This manuscript has been previously reviewed at another journal that is not operating a transparent peer review scheme. This document only contains reviewer comments and rebuttal letters for versions considered at *Nature Communications*.REVIEWER COMMENTS

Reviewer #1 (Remarks to the Author):

I thank the authors for revising their manuscript. This paper announced a public code NMMA and used the code to constrain the neutron star's equation of state by considering the nuclear physics, the gravitational-wave signal, the kilonova, and the GRB afterglow of GW170817, which I found it is interesting. The public code seems also valuable and useful to the community. Here are some suggestions:

(1) In the abstract, "The extension of the NMMA code presented here is the first attempt of analyzing the gravitational-wave signal, the kilonova, and the GRB afterglow simultaneously, which reduces the uncertainty of our constraints." In my previous review report, I clarified that the modeling of GRB afterglow lightcurve can hardly help to constrain the NS EoS. This conclusion is also shown in their results. Thus, it is not true that multi-messenger analysis by including the GRB afterglow can reduce the uncertainty of their constraints. I wish the authors could give a more accurate statement of their results and uncertainties in the abstract.

(2) "It also enables us to classify electromagnetic observations, e.g., to distinguish between supernovae and kilonovae." It is still unclear and confusing to me why supernovae were so important to appear in the abstract, but the authors didn't use them in their manuscript. I only saw them in the method. In my mind, each sentence in the manuscript, especially for the abstract, for a Nature style paper should be valuable. The title, abstract, and manuscript were relevant to binary neutron star mergers and constraints on the NS EoS. Can the observations of supernovae help constrain the EoS? So I didn't know the importance of supernovae here and in this manuscript. I still think the author should simplify or delete unuseful information in their manuscript which may affect readers' reading.

(3) Priors and distributions of parameters should also summarize in Table 2.

Before I recommend this manuscript, I want the authors to address these questions.

Reviewer #2 (Remarks to the Author):

In the revised version, the authors have indeed added some discussion on the uncertainties in kilonova modelling, as well in section 2 as in the conclusions. This discussion remains, however, rather vague, and in particular the assumption of a 1sigma uncertainty on the kilonova lightcurves can in my opinion not replace the inherent uncertainties in the model. One of the main outcomes of the paper is the statement that a simultaneous GW and EM analysis improves noticeably on previous works concerning the radius of a 1.4 solar mass neutron star, $R_{1.4}$. However, this statement depends crucially on the way the uncertainties in the kilonova modeling are incorporated. I understand that at the present stage, it is difficult to include all modeling uncertainties in a quantitative way, but why did they, e.g. not include at least a discussion of the assumption of fixed velocities mentioned by referee #1? Vary at least the assumed 1sigma uncertainty band which is completely ad hoc to see the effect on the extracted value for $R_{1.4}$?

The authors are clearly more careful in their formulations, in particular in the conclusions, but I am still not convinced that all weaknesses of the current state of the work are sufficiently well outlined to recommend publication of this manuscript in Nature Communications.

Max Planck Institute for Gravitational Physics
Multi-messenger Astrophysics of Compact Binaries
Am Mühlenberg 1
14476 Potsdam, Germany

Institute for Physics and Astronomy
Theoretical Astrophysics
Karl-Liebknecht-Str.24-25
14476 Potsdam, Germany

Prof. Dr. Tim Dietrich
Tel. +49 (0) 331-977-230160
Fax +49 (0) 331-977-5617
tim.dietrich@uni-potsdam.de

Assistent: Cornelia Heinrich
Tel. +49 (0) 331-977-5651

Dear referees,

We appreciate your feedback and have modified our manuscript to incorporate all comments.

Reviewer #1

(1) In the abstract, ``The extension of the NMMA code presented here is the first attempt of analyzing the gravitational-wave signal, the kilonova, and the GRB afterglow simultaneously, which reduces the uncertainty of our constraints.'' In my previous review report, I clarified that the modeling of GRB afterglow lightcurve can hardly help to constrain the NS EoS. This conclusion is also shown in their results. Thus, it is not true that multi-messenger analysis by including the GRB afterglow can reduce the uncertainty of their constraints. I wish the authors could give a more accurate statement of their results and uncertainties in the abstract.

For an isotropic kilonova model, indeed, the addition of GRB afterglow within the analysis does not add additional information on the EOS. But for the anisotropic kilonova model, the viewing angle is a non-trivial parameter for the kilonova light curves (e.g. see Fig. 2 in M. Bulla Mon.Not.Roy.Astron.Soc. 520 (2023) 2, 2558-2570), which is correlated with the ejecta mass (e.g. see Fig. 3 in arxiv:2307.11080). Therefore, the additional constraint on the viewing angle from the GRB afterglow propagates to the constraint on the EOS. That also explains the difference between the R14 posterior with and without GRB (Fig. 3), while the GW+KN+GRB result using an isotropic kilonova model (Kasen et. al 2017) is more aligned with the GW+KN result using the kilonova model described in Dietrich et. al (2020).

We consider this statement accurate and crucial for the community, considering that efforts in kilonova modeling are moving towards 2D/3D anisotropic models (e.g. Korobkin O., et al., ApJ 910, 116, 2021, Collins et al., MNRAS 521 1858, 2022, Shingles et al., 2306.17612 2023).

We have modified the abstract so that it simply reads:

“The extension of the NMMA code presented here is the first attempt of analysing the gravitational-wave signal, the kilonova, and the GRB afterglow simultaneously.”

In the main text, we also added:

“Although no significant constraint on the EOS is imposed by the jet energy E_0 as the ratio ξ between it, E_0 , and the disk mass m_{disk} is taken as a free parameter, the inclination constraint from the GRB plays a role in the constraint on EOS. For an anisotropic kilonova model, the inclination angle changes the observable kilonova light curves beyond scaling (e.g. Fig. 4 in Ref.~\cite{Shrestha:2023exe}), which is correlated with the ejecta masses (e.g. see Fig. 3 in Ref.~\cite{Anand:2023jbz}). Therefore the GRB's inclination measurement imposes a constraint on the EOS via the kilonova eject masses measurements.”

(2) “It also enables us to classify electromagnetic observations, e.g., to distinguish between supernovae and kilonovae.” It is still unclear and confusing to me why supernovae were so important to appear in the abstract, but the authors didn't use them in their manuscript. I only saw them in the method. In my mind, each sentence in the manuscript, especially for the abstract, for a Nature style paper should be valuable. The title, abstract, and manuscript were relevant to binary neutron star mergers and constraints on the NS EoS. Can the observations of supernovae help constrain the EoS? So I didn't know the importance of supernovae here and in this manuscript. I still think the author should simplify or delete unuseful information in their manuscript which may affect readers' reading.

We hope that the manuscript under consideration will become the standard reference for the multi-messenger framework NMMA and consider the possibility of performing model selection to be an important part science case. In fact, the code has been used for model selection in Astrophys.J.Lett. 948 (2023) 2, L12; arxiv:2301.02049 (under review in MNRAS); Nature Astron. 5 (2021) 9, 917-927. Note that two of these articles have been submitted during the review of the submitted manuscript.

Nevertheless, we understand that it might be confusing to talk about supernovae in the abstract. Hence we rephrased the abstract:

In previous works, the NMMA code has allowed us to constrain the equation of state of supranuclear dense matter, to measure the Hubble constant, and to compare dense-matter physics probed in neutron-star mergers and in heavy-ion collisions, and to classify electromagnetic observations and perform model selection.

(3) Priors and distributions of parameters should also summarize in Table 2.

We have added the information in Table 2 and also added Table 3, which shows the results for a simulation in which we employ a different kilonova model.

Reviewer #2

In the revised version, the authors have indeed added some discussion on the uncertainties in kilonova modelling, as well in section 2 as in the conclusions. This discussion remains, however, rather vague, and in particular the assumption of a 1sigma uncertainty on the kilonova lightcurves can in my opinion not replace the inherent uncertainties in the model. One of the main outcomes of the paper is the statement that a simultaneous GW and EM analysis improves noticeably on previous works concerning the radius of a 1.4 solar mass neutron star, $R_{1.4}$. However, this statement depends crucially on the way the uncertainties in the kilonova modeling are incorporated. I understand that at the present stage, it is difficult to include all modeling uncertainties in a quantitative way, but why did they, e.g. not include at least a discussion of the assumption of fixed velocities mentioned by referee #1? Vary at least the assumed 1sigma uncertainty band which is completely ad hoc to see the effect on the extracted value for $R_{1.4}$?

We thank the referee for the feedback, and we are grateful that our addition to the text is already considered to be a step in the correct direction.

To further highlight the importance of the uncertainty in the kilonova modeling in our study, we have conducted two sets of kilonova systematic studies.

In the first study, we ran the full GW170817-AT2017gfo-GRB170817A analysis with a different kilonova model. More specifically, the model from Kasen et. al is used. The resulting posteriors can be found in Table 3. The difference in $R_{1.4}$ between this analysis and the main analysis from this work is well within the quoted uncertainty.

Moreover, we have conducted another set of full GW170817-AT2017gfo-GRB170817A analyses with 0.5 mag, 1.0 mag and 2.0 mag uncertainty imposed using our standard model. Running these additional simulations took a significant amount of time, which caused a delay in our reply for which we apologize.

The results from the analysis show that i) the uncertainty on $R_{1.4}$ is the lowest at 1mag, ii) the 0.5mag underestimates the systematics uncertainty while the 2mag overestimates the uncertainty by comparing the best-fit lightcurves with the data. Therefore, we conclude that 1mag is a sensible choice for systematic uncertainty, which is also supported by previous studies performed by the community and cited in the manuscript.

We hope that our additions to the manuscript address the referee's concern.

Tim Dietrich
(for all authors)

REVIEWERS' COMMENTS

Reviewer #1 (Remarks to the Author):

I appreciate the authors' revisions. I have no further comments on this manuscript, and I would like to recommend its acceptance.

Reviewer #2 (Remarks to the Author):

Thanks for the revised version. My concerns have been adressed and I have no further comments.